# Assessment of Processing Parameters of Pack Silicon Cementation onto P265GH Grade Steel

**DOI:** 10.3390/ma16155397

**Published:** 2023-08-01

**Authors:** Mihai Ovidiu Cojocaru, Mihai Branzei, Mircea Dan Morariu

**Affiliations:** 1Department of Metallic Materials Science, Physical Metallurgy, Faculty of Materials Science and Engineering, University POLITEHNICA of Bucharest, 060042 Bucharest, Romania; mihai.cojocaru@upb.ro (M.O.C.); dan.mircea.morariu@gmail.com (M.D.M.); 2Section IX-Materials Science and Engineering, Technical Sciences Academy of Romania, 030167 Bucharest, Romania

**Keywords:** P265GH grade steel, FeSi75C, pack silicon cementation, pack silicon cementation, diffusion porosity, silicide layer growth kinetic

## Abstract

The quantification of the single or combined variation of the process parameters specific to a thermochemical treatment is the key to a full factorial experiment and a first step in the development of computer-aided process engineering. Powdery solid media are frequently used in the practice of thermochemical treatments when the number of processed products is reduced, additional investments are not justified, or when there are no technological alternatives. The control of the process carried out in such powdery solid media involves both the control of the thermal and temporal parameters of the process on the layer growth kinetics and its phase composition, as well as the ratio of the powdery solid mixture mass percentage (active and neutral components, reaction activators, and components with the role of blocking the sintering tendency of the medium particles). In this paper, using the specific full factorial experiment (that is, a first-order complete factorial experiment (CFE)), the full model of the regression equation of the interactions between the specific process parameters of the silicide layer formation in a powdery solid media, applied to the low alloy P265GH steel grade and used in the petrochemical industry, were evaluated. Fe-ARMCO was chosen as the reference in the experimental research carried out.

## 1. Introduction

Silicon cementation in powdery solid media represents, in many situations, a particularly attractive thermochemical processing alternative. The silicon saturation of the silicide layer formation allows a substantial increase in corrosion resistance, refractoriness, hardness, and wear resistance, or, frequently, a combination of these properties imposed by the operating demands of various categories of alloys and parts [1,2,3,4]. In addition, the following categories of such examples can be mentioned: active cemented tungsten carbides mining parts, and compact radiation shielding in fusion power generation [5]; mild steel parts [6,7]; D-type tool steels with high carbon and chromium content [8,9]; 31CrMoV9 grade steel [10]; 304 and 316L stainless steel alloys [11]; carbon/carbon composites (C/C) [12]; and heat-treated low carbon steel (RST 35 grade steel) [13]. The best properties in operation can be seen with parts made of silicon cemented steel where the structure of the layer is represented by a stable ferrite with a high silicon content and without porosity.

The carbon affinity of alloying elements, such as molybdenum and tungsten, increases the thickness of the silicon alloyed ferrite layer.

A particular problem associated with superficial silicon saturation is the appearance of diffusion porosity, which is most commonly related to the difference in partial intrinsic diffusivities of the components, known as the "Kirkendall–Frenkel effect". There is a multitude of solutions to reduce the consequences of the Kirkendall–Frenkel effect’s manifestation [1], but the only guarantee of obtaining a silicide layer without porosity is provided by the absence of the α_1_ phase (the solid solution based on a defined chemical compound of the Fe_3_Si type) [1].

Silicon cementation can also be applied to landmarks made of alloys with a different base than that of iron, for example, aluminum–bronze alloys, providing them with a substantial reduction in the coefficient of friction simultaneously with an increase in wear resistance [14] or alloys with a titanium base [15,16], in order to increase the resistance to oxidation at high temperatures, as well as composite materials [12].

The use of experimental programming methods, and the statistical processing of the data actually obtained by carrying out the imposed cycle of experiments, allows the obtaining of mathematical models of the interactions between the processing parameters (temperature and processing time, the composition of the powder mixture to ensure saturation, etc.) and the growth kinetics of the compound layer [8,9,10,17]. Obtaining such mathematical models is the basis for the optimal management of thermochemical processing: (i) of anticipating the growth kinetics of the compound layer or the appearance of its different phases; (ii) of how the presence of a certain parameter with significant influence can be compensated (when this is required) by modifying the other influencing factors.

## 2. Materials and Methods

The purpose of the experimental research was to quantify the variation of the technological parameters’ effects of the silicon cementation process in a powdery solid media, with ferrosilicon (FeSi75C) as the active component, on the growth kinetics of the total silicized layer and to determine the mathematical models of the interactions between them.

The metallic materials used in the experimental research to highlight the aspects studied were high-purity iron (Fe-ARMCO) and low-alloy grade steel, SR EN ISO 10028-2:2017/P265GH [18], mainly used in the petrochemical industry.

The chemical compositions for the two metallic materials are presented in Table 1, determined spectrometrically using the SPECTROLAB M10/76004135 apparatus (AMETEK, Berwyn, PA, USA). The chemical composition is compared with the mentioned standard, and it is noted that it falls within its limits.

The characteristics of the components of powdered solid mixtures used in the research are as follows:

Ferrosilicon powder (FeSi75C with 72–75% Si and below 0.1% C, with approximately 2% Al) produced by the Norwegian company, FINNFJORD (Finnsnes, Norway); particles with an average equivalent diameter of 40–50 mm were subsequently ground into ball mills to an average the equivalent diameter of 3–4 mm.

Alumina powders (Al_2_O_3_ > 98.5%) were produced at Alum S.A. (Tulcea, Romania), fraction >150 µm max. 10%; fraction < 45 µm max. 12%.

Ammonium chloride (NH_4_Cl) of analytical purity was produced by SilverChemicals (Bucharest, Romania).

The actual thermochemical processing was carried out in electric chamber furnaces, equipped with SiC bars and automatic temperature regulation and control, which were manufactured by UTTIS Industries SRL (Bucharest, Romania).

The samples, with dimensions of 15 × 15 × 10 mm, were packed in solid powder mixtures and positioned in refractory steel containers. After that, the containers were sealed with clay, placed in the furnace at 200 °C, and heated with the furnace up to the holding temperature. After the holding time imposed by the experimental program, the containers were taken out and cooled freely in the air; the extraction of samples was realized when the temperature of the containers was below 300 °C.

The technological parameters of the silicon cemented process in a powdery solid media were: holding temperature at 950, 1050, and 1150 °C; holding time at mentioned temperatures about 1, 3, and 5 h; the proportion of the active component (FeSi75C) of the powdery mixture was 40, 50, and 60%; and the proportion of the ammonium chloride (NH_4_Cl) was 3, 6, and 9%.

The investigation of the obtained results was realized by the use of the following aparatus and techniques:

Microhardness analysis was carried out using a Neophot 21 microscope N1096 series (ZEISS, Oberkochen, Germany) with Hanneman’s indenter under load of 50 gf (HV_0.05_) by means of the Vickers hardness test.

Light Microscopy (LM) was carried out using a Zeiss Z1m Observer microscope—Axio Vision 4.8/038-12837 (Carl Zeiss AG, Oberkochen, Germany).

XRF analyses were performed on the SPECTRO xSORT device no. 143714 (SPECTRO Analytical Instruments, Kleve, Germany), used to verify %Si along the coating thickness layer.

Energy dispersive spectrometry (EDS Sapphire type dispersive energy spectrometer with a resolution of 128 kV) was carried out using a Phenom ProX SEM (Eindhoven, The Netherlands).

## 3. Results and Discussion

The achieving the requirements imposed by the full factorial experiment is represented by the explanation of the mathematical models of the interactions between the independent parameters of the studied process and, respectively, the dependent ones [8,9,10,17].

In the analyzed situation, silicon cementation in a powdery solid media of Fe-ARMCO and low-alloy P265GH grade steel, the independent parameters (inputs factors) X_i_ identified are:

*X*_1_—silicon cementation temperature.

*X*_2_—siliconizing process holding time.

*X*_3_—the proportion of the active component in the mixture—FeSi75C (72–75 ÷ Si, <0.1 %C, about 2% Al).

*X*_4_—the proportion of the activator in the mixture—NH_4_Cl.

The neutral component, with the role of blocking the sintering tendency of the powdery solid media, was Al_2_O_3_, and its proportion was calculated according to the formula [1,2,11,15,16,17,19,20]: %Al_2_O_3_ = 100 − Σ (%FeSi + %NH_4_Cl).

As active components, suppliers of silicon, in the practice of silicon cementation, ferrosilicon powders, silicon, silicon carbide, Si-Ca, or their mixtures are most frequently used in order. As mentioned by Celebi Efe and others [13], when using SiO_2_ as an active component, the presence of pure aluminum powder is also necessary to initiate and maintain the aluminothermic reduction reaction of SiO_2_ and the release of silicon in the active state.

As a dependent parameter, *Y*, the total thickness of the silicide layer obtained for each of the technological variants imposed by the adopted research program was chosen.

Starting from the information given by Lahtin and Arzamasov [2], regarding the limiting role of the penetration of the flow of silicon atoms in the diffusion zone, and, respectively, the similarity between the temperature dependence of the growth kinetics of the compound layer and the silicon diffusion coefficient with temperature, the variant of the full factorial experiment was adopted (first-order complete factorial experiment (CFE) [19,20]). If the statistical checks had not confirmed the correctness of the choice, they would have moved onto a higher-order full factorial experiment.

Following this, we explained the correlations between the independent parameters taken into the analysis (*X*_1_…*X*_4_) and the dependent ones (*Y*) in the form of the full model of the regression equation of the type:(1)$$Y=\delta \left(µm\right)=b_{0}+\sum_{\begin{array}{c} i=1\\ j=1\\ k=1\\ l=1 \end{array}}^{m}b_{i,j\ldots \;}x_{i,\;j\ldots }+\sum_{\begin{array}{c} i=1\\ j=1\\ k=1\\ l=1 \end{array}}^{m}b_{ij;ik;il\ldots \;}x_{i}x_{j}\ldots x_{i}x_{k\ldots }+\sum_{\begin{array}{c} i=1\\ j=1\\ k=1\\ l=1 \end{array}}^{m}b_{ijk;ijl\ldots \;}x_{i}x_{j}x_{k\ldots }$$
where:

*b_i_*; *b_ij_*—the calculated value of Equation (1).

*x_i_*; *x_j_*—the coded values of the factors (independent parameters taken into analysis).

*Y*—the real value of the dependent parameter—total thickness of the silicon cemented layer.

The actual experimental conditions for explaining the regression Equation (1), related to the two situations taken into the analysis (active component FeSi75C) and the results obtained, are presented in Table 2.

The calculation algorithm and statistical verification of the coeficients of the first-order CFE (*b_i_*; *b_ij_*; *b_ijk_*…) involves the determination of the experimental *S*_0_^2^ reproducibility (with the help of the experimental values—minimum 3—obtained under identical conditions), the dispersion in the determination of the *S_bi_*^2^ coefficients, and the confidence interval Δ*b_i_* with which coefficients were calculated; see Table 3 and Table 4.

After the statistical verification of the coefficients and the determination of the particular forms of the calculated full model regression equation, it is necessary to statistically verify their concordance with the help of the dispersion caused by the S^2^_conc_ regression equations and the calculated value of the Fischer criterion (see Table 5).

The comparative analysis of the calculated values with the tabulated ones corresponding to the Fischer criterion (see Table 5), confirms the fact that the calculated regression equations (see Equations (2) and (3)) reflect with maximum probability the singular and collective influences of the independent parameters considered in the analysis on the dependent parameters. Thus, they can be used for process optimization (determining the conditions in which the growth kinetics of the silicide layer reaches the maximum value).

In the case of Fe-ARMCO:*Y* = *δ_t_* [μm] = 166.95 + 75.05*X*_1_ + 50.91*X*_2_ + 32.86*X*_3_ − 16.65*X*_4_ + 40.15*X*_1_*X*_2_ + 22.36*X*_1_*X*_3_ − 13.63*X*_1_*X*_4_ − 36.47*X*_2_*X*_4_ − 42.26*X*_3_*X*_4_ + 40.30*X*_1_*X*_2_*X*_3_ − 50.6*X*_1_*X*_2_*X*_4_ − 40.08*X*_1_*X*_3_*X*_4_(2)

In the case of P265GH grade steel:*Y* = *δ_t_* [μm] = 163.13 + 117.36*X*_1_ + 68.58*X*_2_ + 34.87*X*_3_ + 59.4*X*_1_*X*_2_ + 46.51*X*_1_*X*_3_ + 43.51*X*_1_*X*_4_ + 23.35*X*_2_*X*_3_ − 16.99*X*_2_*X*_4_ + 66.66*X*_1_*X*_2_*X*_3_ + 56.09*X*_2_*X*_3_*X*_4_ + 12.78*X*_1_*X*_2_*X*_3_*X*_4_(3)

The graphical expressions of the regression Equations (2) and (3) made for a series of restrictive packages of variables (for example, the processing temperature and the proportion of NH_4_Cl), as shown in Figure 1 and Figure 2, highlight particularly interesting aspects regarding the effect of the variation independent parameters, for example, of the proportion of the active component and, respectively, of the holding time temperature, on the total thickness of the silicide layer.

The deviations of the values calculated with the regression Equations (2) and (3) in relation to the experimental ones (see Table 2) are reflected by the high dispersion of the experimental values in relation to the calculated ones (experimental reproducibility dispersion) specific to thermochemical processing in powdery solid media; the mathematical models of the analyzed interactions reflect the evolution trends of the formation and growth kinetics of the layers dictated by the variation of the analyzed parameters.

Thus, it is found that, from 1050 °C (see Figure 1 and Figure 2), the increase in the proportion of the active component of the powdery solid media (%FeSi75C) implies an increase in the thickness of the layer, the more energetic and the longer the holding time, and a higher temperature and a lower proportion of the reaction activation component in the media.

High proportions of the component with the role of activator (NH_4_Cl), at the upper limit of the range of values adopted in the active experimental program (i.e., 9%) and associated with temperatures at the lower limit of the same program (950 °C), apparently generate some deviations from the expected mode of variation of the total thickness of the silicide layer, depending on the proportion of the active component in the mixture (%FeSi75C powder) and, respectively, the holding time, as shown in Figure 1 and Figure 2.

Alloying the iron matrix with Mn (1.05% for P265GH grade steel) and increasing the carbon content (0.1% compared to 0.02% for Fe-ARMCO), causes a decrease in the value of the silicon diffusion coefficient in the metal matrix which affects the speed of growth and the formation of the layer, as shown in Figure 3.

Thus, under the same thermochemical processing conditions, the P265GH low-alloy grade steel registers a slower growth kinetics layer than in the case of Fe-ARMCO.

## 4. Discussion

The explanation of such behavior can be found starting from the analysis of probable reactions from a thermodynamic point of view [21] that can take place between the components of the used solid powdery media, and which can ultimately lead to the superficial silicon saturation of the metal matrices.
NH_4_Cl = NH_3_ + HCl(4a)
NH_3_ = N + 3/2H_2_(4b)
(Fe**Si**) + 6HCl = SiCl_4_ + 3H_2_ + **FeCl_2_**(5a)
(Fe**Si**) + N = Fe + Si_2_N   ∆G_T>1000°C_ < 0**↓**Si_2_N^+^ → (Si_2_N^+^)↓ → (Si)↓; (N) ↓ads. ads. ads.(5b)
SiCl_4_ + Fe = SiCl_2_ + **FeCl_2_**(6)
2SiCl_2_ = Si↓ + SiCl_4_ads.(7)
or: Si + 4HCl = SiCl_4_ + 2H_2_(8)
SiCl_4_ + 2Fe = 2**FeCl_2_** + Si↓ads.(9)

The appearance of ferrous chloride (FeCl_2_) in the sequence of reactions, that lead to the formation of silicon in the active state and its proportion, can have undesirable consequences on the kinetics of the cemented layer growth if the proportion of halide in the powdery mixture is not carefully correlated with the processing temperature, respectively, with the proportion of the active component of the mixture.

At temperatures above 900 °C, ferrous chloride is in a liquid state (T_melt_ = 677 °C) and at 1023 °C it starts to vaporize. An excess of it at the media–metal matrix interface, determined by the high proportions of the component with the role of activating the reactions in the media (NH_4_Cl), causes a decrease in the layer formation kinetics due to the blocking of silicon adsorption (an obvious phenomenon in the case of the two investigated metallic materials silicon cemented at 950 °C in media containing 9% NH_4_Cl, as shown in Figure 1 and Figure 2).

Thus, if in the first moments of saturation (a maximum of one hour of holding time), the increase in the proportion of FeSi75C in the powdered solid mixture, in the presence of a high proportion of reaction activator, has an accelerating effect on the kinetics, over time, the accumulation of the ferrous chloride melt makes its effect felt.

Incorrect dosing (overdosing) of the proportion of halide (NH_4_Cl) in the powdered solid mixture when used at low processing temperatures (at the lower limit of the allowed range), can generate major disturbances in the layer growth kinetics, caused by a significant increase in chlorine concentration in the superficial areas of the layer.

X-ray fluorescence investigations carried out on the samples of the two metallic materials silicon cemented at the lower limit of the temperature range, i.e., at 950 °C and at, respectively, high proportions of halide in the powdery solid media (9%NH_4_Cl), highlight the existence of some concentrations’ high levels of chlorine in the superficial areas of the silicide layer along the coating thickness, as shown in Figure 4.

The presence of chlorine in these areas is determined by the accumulation of ferrous chloride at the interface (FeCl_2_ is in a vapor state at the thermochemical processing temperature) and by the pressure exerted by it; changing its state of aggregation upon cooling, it will be found in its crystallized form at room temperature.

With an increase in the thermochemical processing temperature and the proportion of the active component in the powdery solid mixture, and simultaneous decrease in the proportion of the component with the role of activating the reactions between the components of the environment (halides), a decrease in the chlorine concentration at the interface by more than an order of magnitude is recorded (see Figure 4 and Figure 5), as well as a considerable increase in nitrogen concentration. The presence of nitrogen in the silicide layer is determined by the interaction of ferrosilicon with the nitrogen resulting from the ammonia decomposition, an extremely probable interaction (the reaction is exergonic), followed by the ionization of the compound thus produced and the adsorption followed by the decomposition of the Si_2_N^+^ cationic complex, as is shown in Equation (5b). In these new processing conditions, a particular intensification of the growth kinetics of the compound layer is recorded. The mass concentration of nitrogen varies between 1.43% and 1.77%, and of chlorine between 0.00% and 0.06%.

The metallographic analysis performed on samples from the two metallic materials (Fe-ARMCO and P265GH steel grade), with silicon cemented in powdery solid media containing high proportions of FeSi75C and low halides, highlights an area with a columnar aspect specific to the presence of α solid solutions of silicon in iron, and a second one α′, the solid solution based on the Fe_3_Si compound (located outwards), as is shown in Figure 6.

As a particularity, in the case of low-alloy P265GH grade steel, an increase in the proportion of carbon generated by silicon diffusion can be observed in the substrate, which has the effect of a considerable increase in the proportion of pearlite (the structure of these areas corresponds to a hypereutectoid steel).

Another peculiarity of the obtained layers is related to the presence of diffusion porosity, the consequence of the Kirkendall–Frenkel effect.

The microhardness tests carried out on P265GH grade steel samples, thermochemically processed under the mentioned conditions, highlighted the fact that the value in the layer, at about 20 μm of the surface, falls within the range of 150 ÷ 200 μHV_0.05_, regardless of the processing regime, reaching up to about 600 μHV_0.05_ at about 600 μm from the surface. The values represent the average of 10 indentations.

The regression Equations (2) and (3), obtained through the statistical processing of the experimental data, ensure the possibility of more rigorous control of the silicon saturation process:-They allow anticipation by calculation of the silicon cemented thickness layer under strictly specified conditions.-They allow the determination/quantification of the way in which the level of a certain independent parameter can be compensated at a given time by modifying the other influencing factors.-They allow the automatic management of the process.-They allow the construction of graphic representations, as shown in Figure 7, which facilitate the quick determination of how the factors with significant influence must be correlated, within an acceptable error range, to obtain a certain imposed layer thickness.

## 5. Conclusions

Powdery FeSi75C represents a potent active component of powdery solid mixtures used in the silicon saturation process of metal products.The most intense layer growth kinetics is observed with a powdered solid mixture with high values of the active component correlated with low values of the one with the role of activation.The mechanism of silicon adsorption during thermochemical processing has both an atomic and an ionic character.The calculated regression equations of the interactions between the independent parameters of the thermochemical processing and, respectively, the dependent ones (the total thickness of the silicized layer), ensure a multitude of facilities: (i) the optimization of the process; (ii) the estimation of the total thickness of the silicized layer under strictly specified conditions; (iii) the determination of how the deficit or excess of a certain independent parameter can be compensated (if it can be compensated) by changing the others.

## Figures and Tables

**Figure 1 materials-16-05397-f001:**
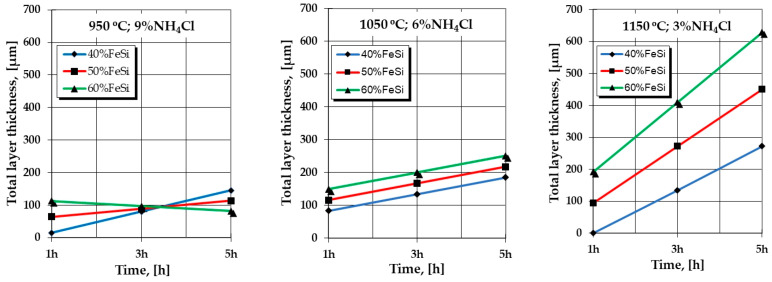
The particular expressions of the regression equations regarding the dependence of the total thickness of the layer (calculated values) on the technological parameters of the silicon cemented process in a powdery solid media, in the case of Fe-ARMCO.

**Figure 2 materials-16-05397-f002:**
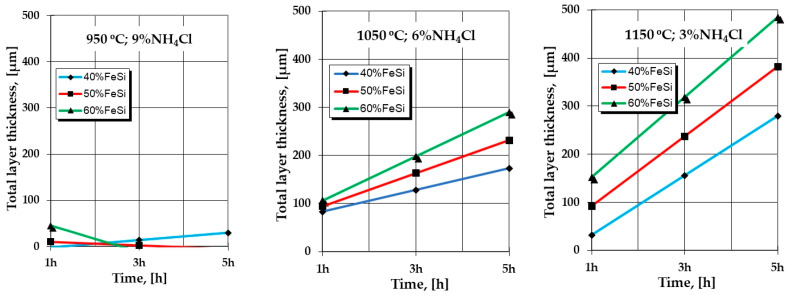
The expressions of the regression equations regarding the dependence of the total thickness of the layer on the process parameters of the silicon cemented process a powdery solid media, in the case of P265GH grade steel.

**Figure 3 materials-16-05397-f003:**
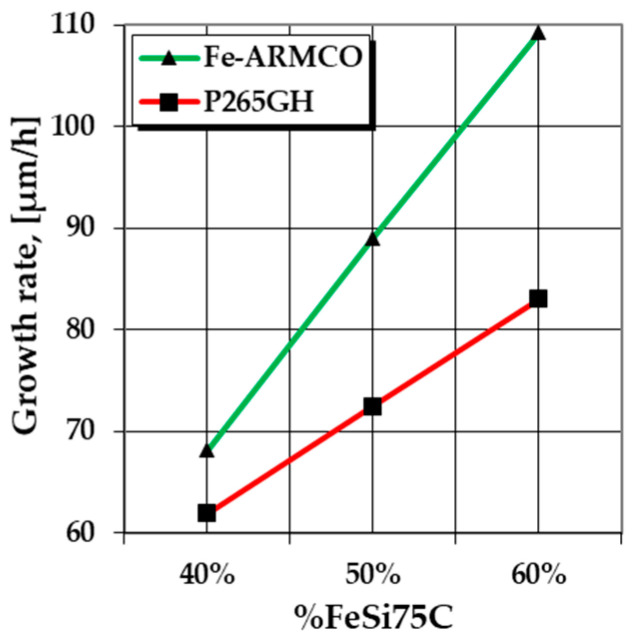
Comparative analysis between the growth rates of the silicide layers depending on the proportion of the active component of the powdery solid media (%FeSi75C), for Fe-ARMCO and P265GH grade steel, at 1150 °C and in the presence of 3%NH_4_Cl.

**Figure 4 materials-16-05397-f004:**
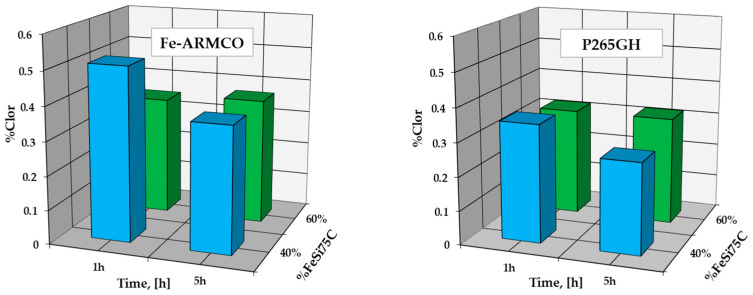
The variation of the chlorine concentration measured in the superficial areas of the silicon cemented layer (950 °C; 9%NH_4_Cl), in powdery solid mixtures containing different proportions of FeSi75C as the active component, for different time periods.

**Figure 5 materials-16-05397-f005:**
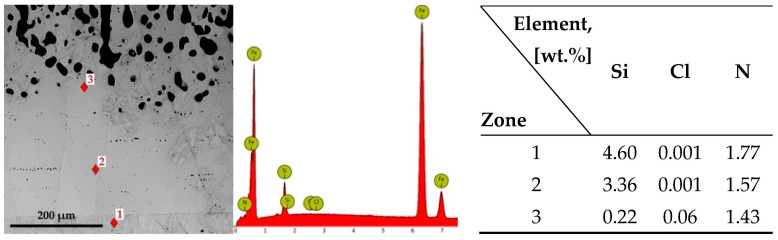
The result of the EDS microanalyses on P265GH grade steel, with silicon cemented at 1150 °C, 5 h holding time, in a powdery solid mixture containing 60% FeSi75C and 3%NH_4_Cl.

**Figure 6 materials-16-05397-f006:**
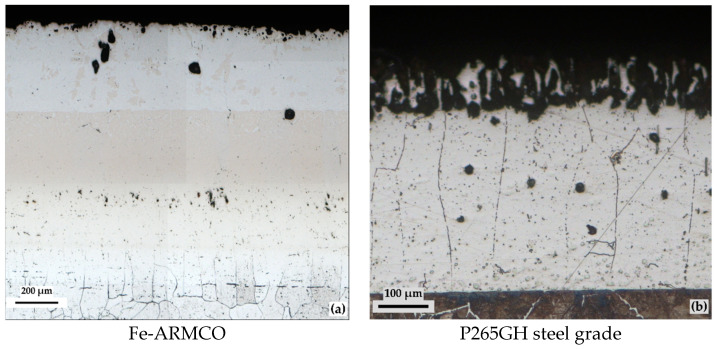
LM of the silicon cemented layers obtained on the two metallic materials processed under conditions that ensure maximum performance (1150 °C/5 h; 60%FeSi75C; 3%NH_4_Cl), for Fe-ARMCO (**a**) and for P265GH grade steel (**b**); NITAL 2% etchant.

**Figure 7 materials-16-05397-f007:**
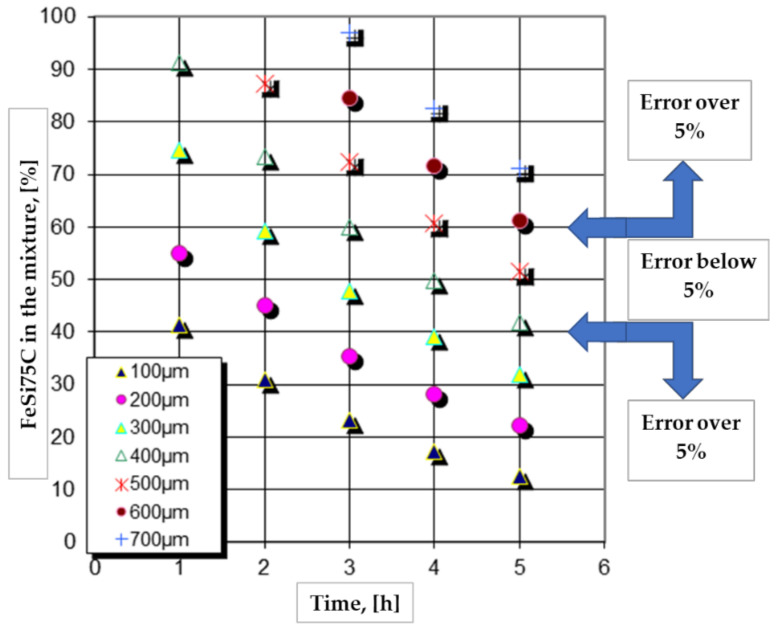
The optimal dosage of the proportion of FeSi75C from the powdery solid mixture used (FeSi75C + 3%NH_4_Cl + Al_2_O_3_) and the holding time at 1150 °C, to ensure a certain total layer thickness, in the case of P265GH grade steel.

**Table 1 materials-16-05397-t001:** The chemical composition of the metallic materials.

%	C	Si	Mn	P	S	Cu	Ni	Cr	Mo	V	Al	Nb	Ti	N	Fe
Fe-ARMCO	0.02	0.07	0.12	0.017	0.011	0.01	0.01	0.003	0.004	0.0005	0.006	-	-	0.008	rest
P265GH	0.10	0.23	1.05	0.014	0.004	0.03	0.02	0.04	0.004	0.002	0.043	0.001	0.002	0.009	rest
SR EN ISO 10028-2:2017	Max 0.20	Max 0.40	0.80–1.40	Max 0.025	Max 0.015	Max 0.30	Max 0.30	Max 0.30	Max 0.08	Max 0.02	Min 0.020	Max 0.020	Max 0.03	Max 0.012	rest

**Table 2 materials-16-05397-t002:** Customization of the 2^4^ (16) first-order complete factorial experiment (CFE).

	*X* _0_	T, [°C] *X*_1_	t, [h] *X*_2_	FeSi75, [%] *X*_3_	NH_4_Cl, [%] *X*_4_	δ_t_—Experimental, [μm]	
						Fe-ARMCO	P265GH
Variation interval (Δ*X_i_*)	Coded variable	100 °C	2h	10%	3%	-	-
High value (*X_i_*_0_ + Δ*X_i_*)		(+1)/1150 °C	(+1)/5 h	(+1) 60%	(+1) 9%	-	-
Standard value (*X_i_*_0_)		(0)/1050 °C	(0)/3 h	(0) 50%	(0) 6%	-	-
Low value (*X*_*i*0_ − Δ*X*_*i*_)		(−1)/950 °C	(−1)/1 h	(−1) 40%	(−1) 3%	-	-
Exp. 1	(+1)	(+1)/1150 °C	(+1)/5 h	(+1) 60%	(+1) 9%	233.76	672.47
Exp. 2	(+1)	(+1)/1150 °C	(+1)/5 h	(+1) 60%	(−1) 3%	612.81	487.29
Exp. 3	(+1)	(+1)/1150 °C	(+1)/5 h	(−1) 40%	(+1) 9%	192.56	177.41
Exp. 4	(+1)	(+1)/1150 °C	(−1)/1 h	(+1) 60%	(+1) 9%	148.96	148.34
Exp. 5	(+1)	(−1)/950 °C	(+1)/5 h	(+1) 60%	(+1) 9%	85.06	0.00
Exp. 6	(+1)	(+1)/1150 °C	(+1)/5 h	(−1) 40%	(−1) 3%	270.94	296.76
Exp. 7	(+1)	(+1)/1150 °C	(−1)/1 h	(+1) 60%	(−1) 3%	171.17	139.42
Exp. 8	(+1)	(+1)/1150 °C	(−1)/1 h	(−1) 40%	(+1) 9%	261.38	288.81
Exp. 9	(+1)	(−1)/950 °C	(−1)/1 h	(+1) 60%	(+1) 9%	137.38	0.00
Exp. 10	(+1)	(−1)/950 °C	(+1)/5 h	(−1) 40%	(+1) 9%	0.00	0.00
Exp. 11	(+1)	(−1)/950 °C	(+1)/5 h	(+1) 60%	(−1) 3%	122.94	0.00
Exp. 12	(+1)	(−1)/950 °C	(−1)/1 h	(−1) 40%	(+1) 9%	0.00	0.00
Exp. 13	(+1)	(−1)/950 °C	(−1)/1 h	(+1) 60%	(−1) 3%	179.44	136.53
Exp. 14	(+1)	(−1)/950 °C	(+1)/5 h	(−1) 40%	(−1) 3%	165.98	219.79
Exp. 15	(+1)	(+1)/1150 °C	(−1)/1 h	(−1) 40%	(−1) 3%	0.00	33.43
Exp. 16	(+1)	(−1)/950 °C	(−1)/1 h	(−1) 40%	(−1) 3%	0.00	9.85

**Table 3 materials-16-05397-t003:** The values of the statistical parameters: *S*_0_^2^, *S_bi_*^2^, and Δ*b_i_*.

Statistical Parameter	Fe-ARMCO		P265GH	
*S* _0_ ^2^	658.26	t_0.05;16_ = 2.12	427.34	t_0.05;16_ = 2.12
*S_bi_*^2^; *S_bij_*^2^	41.14		26.709	
Δ*b_i_;* Δ*b_ij_*…	±13.59		±10.95	

**Table 4 materials-16-05397-t004:** The calculated values of the coefficients of the linear full model regression equation.

Coefficients	Fe-ARMCO		P265GH	
b_0_	166.95	Δ*b_i_*; Δ*b_ij_*… = ±13.59	163.13	Δ*b_i_*; Δ*b_ij_*… = ±10.95
b_1_	75.05		117.36	
b_2_	50.91		68.58	
b_3_	32.86		34.87	
b_4_	−13.65		−2.26	
b_12_	40.15		59.4	
b_13_	22.36		46.51	
b_14_	−13.63		43.51	
b_23_	−12.26		23.35	
b_24_	−36.47		−16.99	
b_34_	−42.26		9.43	
b_123_	40.30		66.66	
b_124_	−50.60		−7.81	
b_234_	9.57		56.09	
b_134_	−40.08		−2.19	
b_1234_	−11.71		12.78	

**Table 5 materials-16-05397-t005:** Statistical verification of the compatibility of the adopted full model equation.

Statistical Parameter	Fe-ARMCO		P265GH	
*F_tabular_*	19.16	V_1_ = 3; V_2_ = 2	19.25	V_1_ = 4; V_2_ = 2
*S* ^2^ * _conc_ *	4775.5		530.16	
*F_calculated_*	7.25		1.24	

## Data Availability

The study did not report any data.

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
