# Peer review of "Assessment of Processing Parameters of Pack Silicon Cementation onto P265GH Grade Steel"

_materials, 2023, doi:10.3390/ma16155397_

Round 1
Reviewer 1 Report
In the paper, using specific experimental programming methods, the solutions of the mathematical models of the interactions between the technical parameters specific to the silicon instrumentation process in a powdery solid media are presented for the case of low alloy steel grade, SR EN ISO 10028-2:2017/P265GH, which is commonly used in the petrochemical industry This job is very interesting and can be accepted with just a few modifications. The comments are as follows:
1. The format of the literature citation does not comply with standards. Please modify the citation label in the main text.
2. Please explain the meaning of the arithmetic expressions appearing in the text, such as by the operating demands of different categories of parts [1 ÷ 13].
3. The direction of the table data section is incorrect, please make modifications, e.g, Table 1
4. Please check the manuscript for grammatical and formal errors,e.g, 172.
5. It is not possible to prove the model and its applicability for process optimization through the analysis in Figure 5 (determining the conditions under which the growth kinetics of the silicon cemented layer is maximal).
6. The attached electron microscope images in the article require a more comprehensive interpretation.
In the paper, using specific experimental programming methods, the solutions of the mathematical models of the interactions between the technical parameters specific to the silicon instrumentation process in a powdery solid media are presented for the case of low alloy steel grade, SR EN ISO 10028-2:2017/P265GH, which is commonly used in the petrochemical industry This job is very interesting and can be accepted with just a few modifications. The comments are as follows:
1. The format of the literature citation does not comply with standards. Please modify the citation label in the main text.
2. Please explain the meaning of the arithmetic expressions appearing in the text, such as by the operating demands of different categories of parts [1 ÷ 13].
3. The direction of the table data section is incorrect, please make modifications, e.g, Table 1
4. Please check the manuscript for grammatical and formal errors,e.g, 172.
5. It is not possible to prove the model and its applicability for process optimization through the analysis in Figure 5 (determining the conditions under which the growth kinetics of the silicon cemented layer is maximal).
6. The attached electron microscope images in the article require a more comprehensive interpretation.
Author Response
Response to Reviewer 1 Comments
Dear Reviewer,
First, I want to thank you again for the patience and pertinence with which you reviewed the paper. The observations/suggestions made are to the point and welcome, both for me and for the journal's level below are presented, in order the changes suggested by you.
- I modified the citation label in the main text..
- I explained/detailed the materials used by different authors in the study of silicon cementation, lines 36-44.
- I modified the direction of the table data (Tables 1, 2, and 4).
- I checked the entire manuscript for grammatical and formal errors, including lines 177-182.
- I modified the meaning of the understanding of the statements, in accordance with your comment 5.
- I modified/completed the text related to Figure 5, corresponding to the SEM-EDS analysis, in accordance with comment 6.
Finally, I want to thank you once again for the time given and for the equidistance with which you analyzed the paper.

Reviewer 2 Report
Silicon Cementation Kinetic in Powdery Solid Media
Abstract:
Line 2: (desideratum) use general words; that could be easily understandable
Line 10-15: rearrange the sentences, as the meaning and concept is not clear.
Line 16: “presupposes a priori” use simple words
Line 15-20: “The process control carried out…tendency of the environmental particle).” Rearrange the sentence, reduce the usage of “and”; disintegrate the sentence.
Line 20-24: “In this paper”, …again the length of sentence is greater; rearrange the sentence.
Line 25-27: the variations…powdery solid media”, rearrange the sentence
Overall, the English and sequence of the content in abstract is not clear.
Abstract need to be reduced too, methodology should not be explained extensively here.
Keywords:
Either write as an alphabetical sequence or may write as in a sequence ARMCO; SR…, diffusion porosity, …
Introduction:
Line 35: in many situation “as” a …
Line 39: reference in wrong format
Line 45: “Kirkendall-Frenkel phenomenon” : the phenomenon is not known by all, so better to clear in simple words.
Line 55-60 and Line 60-65: The sentences are long; rearrange the sentence; even the concept is not clear.
Overall, the introduction is not clear, concept is not clear and language is also poor.
2. Materials and Methods:
Line 80-108: The materials, techniques and methods used should be written in the paragraph, not in bulletin points.
Again the formatting and language is poor.
3 & 4. Results and Discussion
Overall the language is very poor, the results and discussion should be done by indicating the topics/subtopics of each simulation/model.
Moreover, the topics should also be divided with respect to their dependent and independent parameters/variables.
There should be clarity between the experimental values and ones modeled.
5. Conclusions
Line 311: “respectively” what this word shows in the sentence, respective to what?
Could be reduced and here only findings need to be discussed.
Very poor English, require extensive and major revision.
Author Response
Dear Reviewer,
First, I want to thank you again for the patience and pertinence with which you reviewed the manuscript. The observations/suggestions made are to the point and welcome, both for me and the journal's level
Below are presented, in order the changes suggested by you.
- I used general words that could be easily understood.
- I rearranged and reduced the abstract too.
- I rearranged the keywords.
- I modified the reference format.
- In the “Introduction” chapter I explained/detailed the materials used by different authors in the study of silicon cementation (lines 51-60).
- I explained in a few words the "Kirkendall-Frenkel effect" (lines 66-69).
- I rearrange the sentence, as suggested (lines 84-88).
- In the “Materials and Methods” chapter I modified the written in the paragraph (lines 114-121) and I improved the language at the apparatus and techniques descriptions (lines 133-141).
- In the “Results and Discussion” chapter I modified the meaning of the statements' understanding; the agreement between the calculated values and those obtained experimentally (lines 204-209).
- On lines 264-270 you can find the explanation that led to the assumption made, in correlation with the graphs presented in Fig. 2., in addition to the answer from point b), Conclusions.
- On lines 275-288 you can find the explanation regarding point b), from the Concussions.
- I modified/completed the text related to Figure 5, corresponding to the SEM-EDS analysis, by remark a), from the Conclusions (lines 306-322).
- These are the light microscopy images and related comments, which indicate the evolution of the layers according to the experimental conditions, as the response at a) point from the Conclusions (lines 324-340).
Finally, I want to thank you once again for the time given and for the equidistance with which you analyzed the manuscript.

Reviewer 3 Report
In general, the article is interesting, and the information provided by the authors is important.
After the review of the manuscript, I have the following comments:
1. Section 2 Materials and Methods
The experimental conditions of time and temperature are not specified in this section, please include these data.
2. Section 3, Results and discussions
a). No images of the obtained layers appear in the manuscript, probably one or two Figures indicating the evolution of the layers as a function of the experimental conditions could help to understand the experimental results.
b). Line 202,
The authors indicate that “High proportions of the component with the role of activator (NH4Cl), at the upper 202 limit of the range of values adopted in the experimental program (i.e. 9%), associated with 203 temperatures at the lower limit of the same program (950 0C), apparently generate some 204 deviations from the expected mode of variation of the total thickness of the siliconized 205 layer”
However, it is not possible to affirm this assumption because there are not results showing the einteraction of the different temperatures with the same NH4C proportion (3, 6 and 9 %)
Please clarifie that.
Author Response
Dear Reviewer,
First, I want to thank you again for the patience and pertinence with which you reviewed the paper. The observations/suggestions made are to the point and welcome, both for me and for the journal's level.
Below are presented, in order the changes suggested by you.
In section 2 Materials and Methods I introduced the experimental conditions (Lines 131-135).
I excluded the word “respectively” (line 371).
Regarding the English language, I revised the manuscript and corrected the observed mistakes. I have also introduced some additional explanations.
Finally, I want to thank you once again for the time given and for the equidistance with which you analyzed the paper.

Reviewer 4 Report
In the paper the interactions between the technological parameters specific to the silicon cementation process in a powdery solid media are presented for the case of low-alloy steel grade, SR EN ISO 23 10028-2:2017 / P265GH, which is commonly use in the petrochemical industry.
The paper is interesting for the Journal but bout statistical approach it seem a simple application of a standard method provided in previous papers.
About results and comments the contents seem too poor: more comments about experimental results and more links between experimental results and discussion would be needed.
Remarks
The abstract must be rewritten is too long and the first part is not appropriate.
line 55-60 The sentence is long and not clear.
Line 218 "The explanation of such behavior can be found starting from the analysis of probable reactions from a thermodynamic point of view". The following explanation is based on this sentence where a "probable" reaction is reported. Please support discussion with verified results.
Figure 6 Please check caption and provide aption for a) b) and c) subfigures.
Line 287"The microhardness tests" How many tests have been performed? What about position?
Line 293 Equation (2, 3)?
Author Response
Dear Reviewer,
First, I want to thank you again for the patience and pertinence with which you reviewed the manuscript. The observations/suggestions made are to the point and welcome, both for me and for the journal's level
Regarding your remark that the statistical approach is a simple application of a standard method provided in previous papers, the similarities have been generated by the following causes:
In previous papers, the same working tool is used: EXPERIMENTAL PROGRAMMING METHOD – THE ACTIVE.
The experimental conditions are identical but the objectives are totally different.
The statistical processing of the results necessarily follows the same steps and obviously uses the same language.
Below are presented, in order the changes suggested by you.
I rearranged and reduced the abstract too.
I explained/detailed the materials used by different authors in the study of silicon cementation. (Lines 51-60).
I rearrange the sentence, as suggested (lines 84-89 and 92-97).
Regarding the word "probable" (line 254), the explanation is given a few paragraphs below (see lines 264-270).
I modified/completed the text related to Figure 5, corresponding to the SEM-EDS analysis. (Lines 306-322).
Concerning the microhardness test: the position relative to the surface, where the microhardness tests were performed, is already mentioned, and the values represent the average of 10 indentations. (lines 341-345).
I modified "equations (2) and (3)" in the text, as you suggested.
Finally, I want to thank you once again for the time given and for the equidistance with which you ananalyzedhe manuscript.

Round 2
Reviewer 2 Report
Comments
1. The title of the manuscript could be revised for clarity and better coverage too. i.e., Silicon Cementation Kinetic in Powdery Solid Media could be changed to “Assessment of Processing Parameters of Pack Cementation of Silicon onto Steels ” etc., I mean there should be some clarity in the title.
2. Abstract need clarity and English is extremely confusing. For example the word programming was used do not corresponds the computer based applications. Also, the word “Technological parameters” seems to be misleading too. Likewise word concrete was used in Table 2 which again confuses the reader too. This suggests huge English check too.
3. Mathematical model word do not exhibit any coded model used in this study. Did you use any programmer to run the calculations etc.?
4. Kinetic word used in title of the abstract but there is not kinetic study present in this manuscript. Did you use thermoCal or factsage etc. or what thermodynamic software was used in this study?
5. In experimental details, there is no clarity about this word i.e., mixture – FeSi75C. However, it was cleared up in results section. This needs to be taken into account too. Also, the way this word/symbol i.e.,FeSi75C was coined shows that it is an alloy rather a mixture. In addition, the sample was sealed using clay before cementation which seems that there could be huge oxidation. How oxidation of substrate was controlled here?
6. The first thing authors should show is the cross-section of the coating at first which is placed in Figure 6 and secondly there should be XRD of the sample and exhibit the desired structure is formed. Also, Si is not a promising element as for as oxidation is concerned. Why it was selected? In addition, there are three layers in Figure 6, please explain the suitability of each layer, also why there is a lot of porosity? Moreover, the surface topography must be presented too to better understand the surface structure, roughness, size and shape of individual constituents. Also, what is the coating thickness?
7. Moreover, what I can understand from the surface roughness of the Figure 6 right image, that there is huge roughness which is detrimental to both corrosion and oxidation as well.
8. Coating of Si on normal steels is very expensive, yet again why it was selected?
9. Conclusion needs to be made shorter too.
A lot of revision and re-wording is required in entire manuscript.
Author Response
Dear Reviewer,
First, I want to thank you again for the patience and pertinence with which you reviewed the manuscript.
The observations/suggestions made are to the point and welcome, both for me and for the journal's level
Below are presented, in order the changes suggested by you.
- I revised the title of the paper; thank you for your proposal!
- I modified the content of the abstract and reduced it at the same time. I also modified some keywords. I corrected the name of the terms in Table 2.
- As I mentioned in the abstract, I used the specific programming methods full factorial experiment, that is first-order complete factorial experiment (CFE), the full model of the regression equation of the interactions between the specific process parameters of the silicide layer formation in a powdery solid media. Lines 163-168.
- No, I did not use any specialized software, such as Thermo-Calc, but through the active experiment described, I proved the correlation of the independent parameters discussed, with the resulting layer thickness, Fig. 7 being suggestive in this regard.
- The characteristics of the components of powdered solid mixtures used in the research, including FeSi75C, are presented in lines 80-89.
- The answers to point 6 can be found in the text at: lines 47-51; lines 276-292; lines 293-309; lines 330-314; lines 329-332.
- Regarding the oxidation, indeed it is accentuated, resulting in a layer like the one in Fig. 6; any way he is approx. 4-5 times thinner than the silicized layer itself. After the thermochemical treatment, this layer is removed mechanically, so the properties like corrosion and oxidation as well, will not be affected.
- As I showed in the introduction, supported by 13 references, silicon cementation is the most convenient option for improving corrosion and oxidation resistance in environments specific to the petrochemical industry. Lines 31-42.
- I made the conclusion shorter; see lines 334-346.
Finally, I want to thank you once again for the time given and for the equidistance with which you analyzed the manuscript.

Reviewer 4 Report
Thanks to the author for the review of the paper
Author Response
Thanks to the reviewer for the review of the paper